# Systematic Review and Meta-Analysis: Identification of Factors Influencing Milking Frequency of Cows in Automatic Milking Systems Combined with Grazing

**DOI:** 10.3390/ani10050913

**Published:** 2020-05-25

**Authors:** Françoise Lessire, Nassim Moula, Jean-Luc Hornick, Isabelle Dufrasne

**Affiliations:** 1Centre des Technologies Agronomiques de Strée, rue de la Charmille, 4577 Strée, Belgium; projet.formation@cta-stree.be; 2Department of Veterinary Management of Animal Resources, Faculty of Veterinary Medicine, Fundamental and Applied Research for Animal and Health (FARAH), University of Liège, 4000 Liège, Belgium; nassim.moula@uliege.be (N.M.); jlhornick@ulg.ac.be (J.-L.H.)

**Keywords:** automatic milking system, grazing, pasture-based automatic milking system, AMS, milking frequency, meta-analysis

## Abstract

**Simple Summary:**

Numerous publications have investigated the possibility of combining automatic milking systems (AMS) with grazing. Milking frequency (MF) was usually considered as an indicator of robot performance and researchers focused on ways to optimize it. It seems pertinent to compile the published results. By using principal component analysis, we discriminated four agricultural exploitation systems (clusters). These systems differed from low—less than two milkings per day—to MF similar to those recorded at barn (2.7 milkings/cow per day). The description of clusters allowed for the identification of parameters influencing MF: concentrate supply, minimum milking interval, pasture dry matter intake and stage of lactation. By pair-wise analysis, we quantified the relationship between each parameter and MF. In a second step, we identified the relationship between MF and milk production (MY). These analyses allowed us to understand in which context these parameters could be efficient. For example, concentrate supply in full grazing has limited efficiency but in early lactation, increases MF. High percentage of grazed grass in a cow’s diet seems to limit MF. The impact of MF on MY was confirmed. In conclusion, several strategies can be implemented to combine grazing and AMS with an impact on productivity and on production costs.

**Abstract:**

More dairy farms (up to more than one in four in some countries) are equipped with automatic milking systems (AMS) worldwide. Because of the positive impacts of grazing, e.g., on animal welfare or on production costs, numerous researchers have published papers on the combination of AMS with grazing. However, pasture-based AMS usually causes a reduction in milking frequency (MF) compared to indoors systems. The objectives of this meta-analysis were to review publications on the impacts of pasture-based AMS on MF and mitigation strategies. First, data from 43 selected studies were gathered in a dataset including 14 parameters, and on which a Principal Component Analysis (PCA) was performed, leading to the description of four clusters summarizing different management practices. Multiple pairwise comparisons were performed to determine the relationship between the highlighted parameters of MF on milk yield (MY). From these different analyses, the relationship between MF and MY was confirmed, the systems, i.e., Clusters 1 and 2, that experienced the lowest MF also demonstrated the lowest MY/cow per day. In these clusters, grazed grass was an essential component of the cow’s diet and low feeding costs compensated MY reduction. The management options described in Clusters 3 and 4 allowed maintenance of MF and MY by complementing the cows’ diets with concentrates or partial mixed ration supplied at the AMS feeding bin or provided at barn. The chosen management options were closely linked to the geographical origin of the papers indicating that other factors (e.g., climatic conditions or available grasslands) could be decisional key points for AMS management strategies.

## 1. Introduction

The expansion of robotic milking is exponential. Around 25,000 automatic milking systems (AMS) were installed worldwide from 2011 to 2014 [1]. In Europe, this trend is even more marked: about 25% of dairy farms in Denmark and about 20% in Sweden were equipped with a robot in 2014 [2]. However, the automation of milking is too often linked to a decrease in utilisation of grazing [3,4]. Grazing offers many advantages, including improving animal welfare [5], decreasing feeding costs [6,7] and is beneficial in some ways for the environment. It also has a good image for the consumers [8]. Many publications demonstrated that combining robotic milking and grazing was possible by using strategies based on a diet of exclusively grazed grass or on maximum milk yield. Different layouts can be adapted to find a compromise between these two options. Pasture-based AMS installed on pastures with free access to paddocks 24 h/day allow for the maximum use of grass. These systems rely almost exclusively on grass. Conversely, AMS within the barn could be designed to allow the cows access—under direction or not—to the pasture for a variable time depending on the system. In this case, grazed grass is considered a complement of the partial mixed ration (PMR) given at barn that allows maintenance of a high milk yield level. Whatever the developed strategies, the reviewed literature considers that traffic to the robot is an important parameter to evaluate the productivity of the system. Several authors report that AMS associated with grazing reduces milking frequency (MF: number of milkings/cow per day) in comparison to indoor AMS, while this effect is restricted with indoor robots [9,10] allowing for access to pastures. In this context, the objective of this study was to identify the different factors influencing MF in pasture-based AMS. Then, a meta-analytic approach was conducted in two steps: the first aim was to characterise the different systems described by the literature through principal component analysis (PCA). The description of different management strategies allows us to estimate the feeding costs of each cluster and to assess whether the decrease in these costs, at cow level, could compensate for the lower income associated with a low milk production. In the second step, based on PCA results, a pairwise comparison was conducted to identify the factors influencing MF. Finally, the relationship between MF and milk yield (MY) was evaluated. The pertinence of using MF as proxy indicator of the productivity and consequently of the profitability was thus questioned.

## 2. Materials and Methods

### 2.1. Literature Search and Strategy

The systematic review was conducted following the Prisma statement defined as an evidence-based minimum set of items for reporting in systematic reviews and meta-analyses [11]. The literature search started in July 2018 with the use of different databases: Scopus, Science Direct and MedLine using the MeSH keywords. It ended in December 2018. The following terms selected on basis PICOS statement were combined: automat* milking system OR robot*, cattle, pastur* OR grazing, milk* frequency OR interval and traffic OR voluntary. They had to be included in the title, keywords or abstract. Thereafter, we checked the references of the selected reviews to verify whether they could be possibly included in this systematic review. Two reviewers examined the papers using a standardized procedure including selection criteria discussed between them prior to the articles’ inclusion. Conflicts were submitted for advice to a third reviewer. After this step, 71 articles and reviews were kept for further analysis.

### 2.2. Eligibility Criteria

The following criteria determined the inclusion after the first screening: 

Only papers written in English or French were accepted. Other languages were excluded. The papers had to have been published between the years 2000 and 2018. Studies, reviews and conference papers were included, but not books. We made the choice to exclude rotary milking systems (RMS). In actuality, concentrates are not delivered in the feeding bin of the RMS, but in feeding stalls installed outside of the platform and therefore could influence traffic of cows to the AMS. A second evaluation was then conducted following these conditions of acceptance: the study had to mention the effect of AMS with access to grass on MF or milking interval, quantitative information of MF had to be delivered in the paper and when missing information, authors were contacted and invited to complete it. After the second screening, 43 papers from 71 were selected.

### 2.3. Descriptive Synthesis

The geographical area from the 43 studies was identified and could be divided into quarters: the first quarter consisted of Australian studies (27%) describing mixed systems combining forages and a grazed grass allocation on pasture-based robots, the second consisted of studies performed in New Zealand and Ireland, respectively (16% and 11% respectively) where grass was a major constituent of cows’ rations, the next quarter involved Belgian and American papers (13% and 11% respectively) with limited grazing seasons, and finally the last one included studies from France and Northern European countries with restricted daily access to pastures. The geographical origin of data from reviews was not taken into consideration in these results.

The type of publications was also analysed: proceedings (15) and research papers (20) constituted the main proportion of the included papers. The remainder comprised of three short communications and 5 reviews. These reviews were checked to assess that all the papers they referred to were included in the evaluation process. Only results from the refereed papers were included in the datasets.

Only 2 papers concerned organic farming [12,13]. Some proceedings publications included the description of systems developed in experimental farms, for example in Derval or Trevarez in France, DairyNZ Greenfield farm in New Zealand (NZ), Camden in Australia or W.K Kellog Biological station in USA. Such publications aimed to give an overview of their practices and outcomes with, sometimes, incomplete statistical indicators (standard deviation or SD, standard error or SE, *p*-value). However, the provided data based on several years of observations was considered very relevant. We therefore agreed to include it in the first dataset. The second dataset accepted studies with indicators allowing the results to be objectively weighted into consideration. 

The factors that influenced MF were listed in the included studies and classified as suggested by Lessire et al. (2013) [14] in 2 categories: parameters that are manageable and parameters that are non-manageable by the farmer. The farmer can solve the manageable issues by adapting the robot management (RM): e.g., the parameters of the AMS (minimum milking interval (h)—MMI, number of cows per robot (n cows), concentrate supply (CS—kg/cow per day) in the robot’s feeding bin and feed complementation at barn based on harvested forages or PMR (kg DM) or the grazing management (GM), e.g., pasture dry matter intake (PDMI—kg DM), pasture allocation, sward height, stocking rate, water availability and distance to the robot). The factors linked to the animal (hierarchy, social behaviour, stage of lactation, parity, breed, health condition) can be manageable to some extent and so were considered as manageable (Herd management: HM). Finally, climate conditions were pointed out as the sole non-manageable conditions. In Table 1, we present a summary of the reviews selected in this study. In Table 2, we display an overview of all the included papers classified as defined before and a categorical description of the studied parameter on MF (reported increasing or lowering effect). 

### 2.4. Meta-Analysis

The 43 studies were classified on the basis of the highlighted factors. A first dataset only involving studies provided precise information about PDMI compiled data on 14 parameters, 10 quantitative and 4 categorical ones. Four parameters were directly related to RM: number of cows per AMS, CS, minimum milking interval (i.e., the minimum time elapsed between 2 milkings for a new access—MMI) and the supply of eventual complements, i.e., harvested forages or partial mixed ration (PMR). Three parameters of GM were recorded: maximum distance between the robot and pastures, stocking rate and PDMI. Milk yield and MF gave an overview of robot performance. The duration of the study was indicated. The categorical parameters were the type of grazing management (rotational, rotational simplified and strip grazing), the pasture allocation (A, AB, ABC with respective access to one block of pastures per day, day and night access to different blocks and finally access to 3 blocks per day, the type of complement (no complement, harvested forages, PMR). In order to fulfil a dataset including a significant number of publications, data about nutritional values or botanical composition were not included. Geographical origin was noted. When needed, we converted data to use the same unit (e.g., milk yield converted from litres to kilos). In some cases, the diet of the cows was composed of grazed grass, concentrate and complement, but did not feature pasture dry matter intake. In this particular context, the daily dry matter intake (DMI) was estimated following the equation developed by the National Research Council [50]. Thus, PDMI was calculated by subtracting the amount of concentrates and complement from this value. We deduced some missing data about the farm layout from the previous studies based on the same institution: e.g., the area and the pasture allocation were considered as the same in the different studies conducted at the same time (e.g., data from the experimental station in Camden in New South Wales (Australia)). To supplement missing data, we requested help from authors, but some figures had to be identified as not available (NA). Non-available data was mainly linked to MMI (13 NA/54) and to the amount of complement (8/54), concentrate (5/54) and grass (5/54) in the diet of the animals. 

### 2.5. Impact of Developed Strategies on Feeding Costs

To evaluate the strategies developed in the 4 clusters from an economic point of view and to calculate the feeding costs, we introduced the following prices of feedstuffs based on their application in the country where the management described by the Cluster was the most represented. For Cluster 1, we applied feeding costs used in several studies [51,52], i.e., production cost of grazed grass evaluated at 0.08 €/kg DM and concentrate price at 0.250 €/kg DM. For Clusters 2 and 3, we chose Belgium as a reference: production cost of grazed grass was evaluated at 0.08 €/kg DM, price for purchase of concentrate (16% CP—crude protein) and PMR was evaluated at 0.285 €/kg DM and 0.221 €/kg DM, respectively [53]. The calculation of feeding costs in Cluster 4 was made based on PMR and concentrates feeding from the United States published by Gillespie (2019) [54] and Saint Pierre (2019) [55] with the production cost of grazed grass evaluated at 0.07 €/kg DM, price for purchased concentrate (16% CP—crude protein) and PMR at 0.275 €/kg DM and 0.229 €/kg DM, respectively.

### 2.6. Statistical Analysis

Statistical analyses of the dataset were performed using different software: the software R (R-core Team 2016), SAS (9.3) and Revman5.3 [56]. Descriptive analysis was carried out first, followed by analysis in Multivariate PCA using the package FactorMine R functions PCA. The hierarchical classification (HC) (proc CLUSTER) using Wald’s algorithm was then performed to achieve the determination of groups with common characteristics. A GLM (Generalized Linear Model) procedure was used to investigate which continuous factors were significantly different from one cluster to the other. The Proc Freq procedure was used to determine the impact of categorical factors. The significance was assessed by the method of χ^2^ of Mantel–Haenszel.

This first evaluation allowed us to investigate the relationship between MF and the 14 identified parameters. Correlation coefficients between MF and the different parameters were calculated. Factors influencing MF (concentrate and PDMI, stage of lactation, MMI) and included in the clusters’ descriptions were further analysed by multiple pairwise comparisons using RevMan5.3. At this stage, only studies mentioning SD, SE, Standard Error of the Difference (SED) or Standard Error of the Mean (SEM) were included. The means and SD of each study were introduced and weighted. The mean difference and the standard error of the difference between 2 outcomes were calculated for each study. Forest plots allowed for the visual assessment of heterogeneity. Statistical tests were used to ensure its objectivity. The χ^2^ test was used and the null hypothesis (homogeneity of outcomes) was rejected at *p*-value < 0.05. The inconsistency between studies was measured by I² determined by the ratio of (χ^2^ – df) on χ^2^. Below 50%, heterogeneity was considered as low, whereas above 50%, it was considered as high. At high heterogeneity, we used random effect models to consider the variability of settings between studies [57,58]. Subgroups were constituted to identify factors leading to heterogeneity. A test for overall effect was estimated by z-test and its *p*-value [56].

## 3. Results

### 3.1. Analysis in Principal Component

Table 3 presents the results of the descriptive statistical analysis of the quantitative variables. Fourteen variables were analysed by PCA.

#### 3.1.1. Study of Outliers

The Figure 1 and Figure 2 show the graphic representation of all the papers included in the dataset and identified by ID, on axis 1 and 2 and axis 2 and 3, respectively. As shown on Figure 1, all the studies were grouped except for one from NZ (ID: 31—Cluster 1) and one from the USA (ID: 43—Cluster 4).

#### 3.1.2. Inertia Analysis

Axis 1 and 2 represented 41% and 18.8% of the total variation, respectively (Figure 1). Axis 3 achieved 13.1% (Figure 2). Thus 72.9% of the variation could be explained by the circle of correlation (Figure 3) between variables studied and axes. Other axes’ participation was less significant and added only minor indications. The Hierarchical Cluster Analysis (HCA) of the studied articles defined four clusters.

#### 3.1.3. Correlation Circle Representation

The correlations of the quantitative variables with the first and second axes are shown in Figure 3. The results indicated a positive correlation of MF, CS, MY and complement with Axis 1 (correlation values = 0.90, 0.83, 0.79 and 0.63, respectively). A negative correlation with this axis was observed for the number of cows per AMS (−0.60), PDMI (−0.65) and stocking rate (−0.56). We can thus consider that this first axis characterized intensive systems with high production levels. Positive correlations with axis 2 were reported for complement (0.60), trial duration (0.60) and number of cows per AMS (0.57) while negative values were allocated to PDMI (−0.62). This axis could describe mixed systems relying more on complement allocation than on grass. The third axis was mainly correlated with maximum distance to the robot (0.723) and (negatively) with MMI (−0.656), both factors being manageable factors (RM and GM).

#### 3.1.4. Impact of Developed Strategies on Feeding Costs

The strategies developed in the four clusters were evaluated from an economic point of view. By using economic data described in the Material and Methods, the feeding cost was estimated at 0.11 €, 0.11 €, 0.13 € and 0.15 €/kg milk in Cluster 1 to 4, respectively. We included the mean MY (kg/cow per day) and the number of cows per robot from the cluster description to determine the total feeding costs per AMS. The total MY was 1260 kg, 1046 kg, 1809 kg and 1387 kg/AMS in Cluster 1 to 4, respectively. Feeding costs per AMS averaged thus 138.6 €, 115.1 €, 235 € and 210.3 €/AMS for Cluster 1 to 4, respectively.

### 3.2. Multiple Pairwise Comparison

The correlation coefficients between MF and the most relevant variables highlighted by the PCA analysis and by the literature review helped us to choose parameters (common determinators) to be included in multiple pairwise analysis. The selected common determinators were the CS, MMI, Stage of Lactation (SOL) and PDMI. Only papers providing statistical indicators such as SD, SE, SEM or SED were included in this analysis. RevMan5.3 was then used to estimate the heterogeneity of the pooled studies with eventual subgroups analysed when heterogeneity was observed. The statistical indicators (Relative weight of studies, τ^2^, χ^2^, I^2^ and *p* and z-value) are indicated on Figure 4, Figure 5, Figure 6, Figure 7, Figure 8 and Figure 9 and thus are not detailed in the description of the Results section.

#### 3.2.1. Effect of the Concentrate Supply on Milking Frequency

We selected this determinator because of its strong correlation with axis 1 in PCA. Furthermore, numerous papers included in the systematic review considered that the supply of high amounts of concentrate in the feeding bin of the AMS is an incentive to encourage cow traffic to the AMS [59,60]. According to the systematic review, we hypothesized that high concentrate supply (HC) increased the MF while low CS (LC) decreased it. The dataset was completed with a total number of five pairs of results pooled for multiple pairwise comparisons. A minimum delta of 2 kg/cow per day was necessary for this comparison. The result of this analysis demonstrated that the increase in CS by 2 kg/cow per day induced an increase in MF of 0.12 milkings/cow per day (Confidence intervals 95% (CI) [−0.05; 0.29]; 356 data; *p* = 0.16). The forest plot (Figure 4) showed divergent results so we constituted subgroups. Five effects (stage of lactation, MMI (≥6 h), MMI (4 h), breed, number of lactations identified in Figure 4) were investigated. The effect of HC on MF was compared with LC for cows (first comparison) in early and then in late SOL in the first subgroup (1.1.1). The effect was less than expected considering the global analysis: CS induced an increase of 0.03 milkings/cow per day; CI [−0.03, 0.09] and was statistically not significant (*p* = 0.35).

Two modalities of MMI—MMI ≥ 6 h (1.1.2) and MMI = 4 h (1.1.3)—were studied in two other subgroups (1.1.2 and 1.1.3). The comparison of HC X MMI 6 h vs. LC X MMI 6 h demonstrated a rise in MF of 0.43, CI [0.12, 0.74]; *p* = 0.006). For subgroup 1.1.3, no significant effect of CS was observed. For subgroup 1.1.4, data were collected from Nieman et al., 2015 [40] where high CS delivered to the Holstein breed was compared with low CS delivered to New Zealand Friesian cows. No significant effect (*p* = 0.55) was recorded. In subgroup 1.1.5, the effect of CS was studied in two groups of multiparous cows (high CS vs. low CS). A significant decrease in MF was observed and estimated at −0.14 milkings/cow per day (CI [−0.22; −0.06]). Regarding the results of these pairwise comparisons, only HC combined with MMI ≥ 6 h was effective at increasing MF.

We studied the effect of concentrate allocation in the different clusters defined in the PCA analysis (Figure 5): three subgroups were formed with three studies in Cluster 1, five studies in Cluster 2 and one study in Cluster 4. Milking frequency was significantly increased by 0.09 milkings/cow per day (CI [0.06, 0.13]—*p* < 0.00001) and by 0.08 milkings/cow per day (CI [0.01; 0.16]; *p* = 0.03) in Clusters 1 and 2, respectively. The effect of CS in Cluster 4 was significant (*p* < 0.0001) and the estimated effect was more important than in other clusters (increase of 0.49 milkings/cow per day; CI [0.25; 0.73]). Overall, the significant effect was lower than in the previous comparison: an increase of 0.10 milkings/cow per day was linked to HC (CI [0.04; 0.15], *p* < 0.0001). This estimation arose mainly from Clusters 1 and 2 as the relative weight of Cluster 4 was low (4.3%).

#### 3.2.2. Effect of the Minimum Milking Interval on Milking Frequency

We analyzed the influence of this parameter, as it seems to interfere with the impact of CS. The prerequisite was that giving more opportunities to cows to be milked (i.e., short MMI that reduces the time necessary to be admitted for a new milking) would increase MF. Four studies specifically studying this impact were included in the pooled dataset. Milking frequencies related to short MMI were as opposed to those with extended MMI (Figure 6). The mean of the difference was positive indicating a positive effect of the determinator: the short MMI increased the milking frequency by 0.37 milkings/cow per day; CI [0.20; 0.54], *p* < 0.0001.

In the subgroup 3.1.1, MMI set at 8 h (Foley et al., 2015_b_) [24] or 6 h (Jago et al., 2004 ) [23] was challenged to MMI of 12 h (in both studies) while LC was supplied (≤ 1 kg/cow per day). A significant effect was highlighted with an increase in MF estimated at 0.37 (CI [0.19; 0.55]). In the subgroup 3.1.2, MMI 4 h was confronted to MMI of 6 h. The difference between the treatments of 0.11 h, (CI [0.03; 0.19], *p* < 0.008) was significant. In the subgroup 3.1.3, changing MMI from 12 h to 7.5 h induced a significant increase in MF estimated at 0.60 milking/cow per day; CI [0.56; 0.64]. To summarize these pairwise comparisons, short MMI induced an increase in MF and this increase was more marked for MMI changing from 12 h to 7.5 h with or without low CS.

#### 3.2.3. Effect of the Stage of Lactation on Milking Frequency.

The effect of SOL was investigated (Figure 7). Two groups were considered: one with cows in early lactation, i.e., less than 100 d in milk (DIM) compared to late lactation with DIM > 200 d. The global analysis of SOL showed a positive effect: early lactation cows were milked more frequently. The increase in MF was estimated at 0.34 milkings/cow per day in early lactating cows compared to late ones CI [0.16; 0.52], *p* = 0.0003. However, the observed heterogeneity prompted us to form subgroups based on similar experimental designs. In the subgroup 4.1.1 (High delta), we noted that the cows in early lactation received on average 1.82 kg of concentrate/cow per day more than the late lactation cows. In the subgroup 4.1.2 (Low delta), the cows with DIM < 100 d received 5.29 kg of concentrate/cow per day and those with DIM > 200 d, 4.65 kg/cow per day. In this subgroup, the increase in MF in the early stage of lactation was not significant and estimated at 0.02 milkings/cow per day; CI [-0.07; 0.11] for early lactation cows while in subgroup 4.1.1 (High delta CS), the increase in MF was significant and evaluated at 0.42 milkings/cow per day; CI [0.31; 0.52]. In summary, an early lactation stage caused increase in MF but its magnitude depended on CS.

#### 3.2.4. Effect of Pasture Dry Matter Intake on Milking Frequency

Two studies investigated the effect of high or low grass allocation on MF. They were pooled to investigate the effect of PDMI on the MF (Figure 8). PDMI over 17 kg DM, considered as high, caused a significant decrease in MF by 0.30 milkings/cow per day; CI [−0.49; −0.10].

#### 3.2.5. Effect of Milking Frequency on Milk Yield

After these different analyses, we chose to challenge the hypothesis that the higher the MF, the higher milk yield. We considered study results arising from comparison of large differences of MF, i.e., from 0.3 milkings/cow per day (Figure 9). The effect range was a gain of 4.70 kg milk/cow per day; CI [2.44; 6.96] for an increase of about 0.3 of the milking frequencies. All the included studies showed an increase in MY when MF was higher.

To investigate the factors leading to heterogeneity, we formed six subgroups. The subgroup 6.1.1—SOL at low CS (less than 1 kg/cow per day)—showed a very marked outcome. In that group, the difference in MY observed between cows in early and late lactation reached 11.91 kg/cow per day, CI [9.66; 14.16], *p* < 0.00001. Other subgroups leading to significant effects were the following: effect of rumination, of pasture allocation and of stocking rate. The subgroup “effect of rumination” (6.1.2) compared high ruminating cows vs. low ruminating ones with an increase in MF of 14.02 kg/cow per day, CI [9.36; 18.68], *p* < 0.00001 for high ruminating cows. The subgroup 6.1.3 investigated the effect of SOL X pasture allocation. The article coupled change in pasture design (ABC vs. AB) with SOL (early, mid and late) (Lyons et al., 2013c) [34]. The pasture design ABC corresponded with total paddock area divided into three blocks, each of them being accessible during 8 h. The AB design corresponded to the division of the total paddocks in two blocks, one being accessible during the night (12 h), the other during the day (12 h). Access was provided through smart-gates. Pasture allocation ABC vs. AB reached 3.92 kg milk/cows per day CI [2.97; 4.87], *p* < 0.00001. The subgroup 6.1.4 studied the effect of stocking rate (High stocking rate = three cows/ha vs. low stocking rate = two cows/ha). The comparison demonstrated an increase in MY at increasing MF estimated at 2.30 kg/cow per day, CI [0.51; 4.09], *p* = 0.01. The last two subgroups included comparisons of high dominant vs. low dominant cows with or without limited access to water (6.1.5) and in subgroup 6.1.6, the effect of short (6 or 7.5 h) vs. long MMI (12 h) both of which showed no significant effect. In summary, when increased MF induced increased MY, this effect was more marked in early lactation stage and in ABC pasture design.

## 4. Discussion

The first objective of this study was to identify factors influencing MF in pasture-based AMS. First, we compared the data provided by the selected studies to obtain a general overview of management systems described in the literature. The PCA was useful for the discrimination of four different systems identified by clusters. Milking frequency increased from Cluster 1 to Cluster 4 and lead to an increase in MY: from less than 2 milkings/cow per day (Cluster 1) with MY < 20 kg/cow per day to 2.22 milkings/cow per day with an average MY of 27 kg/cow per day (Cluster 3) and finally 2.63 milkings/cow per day and an average MY of 28.9 kg/cow per day. A deeper evaluation showed that the increase in MY was correlated with increased CS in the feeding bin and PMR delivered at barn. Yet, in Cluster 4, high MY (MY of 28.9 ± 1.2 kg/cow per day) combined with high MF (MF of 2.63 milkings/cow per day) were reached with a diet including CS of 6.56 ± 0.42 kg/cow per day in the feeding bin of the AMS and PMR (8.4 ± 1.4 kg DM). Grazed grass in cows’ diets was thus restricted to about 50% of DMI in Cluster 3 and 30% in Cluster 4. Thus, the total ration (PMR + concentrate + grazed grass) given to cows seems more determinant than CS + grazed grass to explain the increase in MF and MY.

Grazed grass was an essential component representing 90% of the cows’ diets in Clusters 1 and 2 with no or little complement supplied. These clusters are differentiated by three parameters linked to robot management. The occupation rate of AMS was lowest in Cluster 2 than in Cluster 1 (51.3 cows per AMS in Cluster 2 vs. 75.1 cows per AMS in Cluster 1), the CS was enhanced (from 1.92 in Cluster 1 to 2.61 kg/cow per day in Cluster 2) and the MMI was shortened from 11.0 ± 0.6 h in Cluster 1 to 6.3 ± 0.6 in Cluster 2. These changes in parameters induced an-increase in MY by about 4 kg/cow per day and an increase in MF of 0.6 units in Cluster 2 compared with Cluster 1.

These contrasted systems described in clusters were strongly correlated with geographical origins of papers. It is not surprising, considering the environmental and agricultural constraints of the respective countries. Ireland and New Zealand belong to Cluster 1. In actuality, the high percentage of grazed grass in these countries is related to the huge percentage of agricultural area dedicated to grassland and grazing. In fact, 81% of Irish agricultural area is devoted to pastures, hay and grass silage production [61]. In New Zealand, 2.6 million ha of grassland are used for dairy production [18,62] and grazed grass constituted 82% of cows’ diets [63]. Belgian studies are mainly represented in Cluster 2 as grazing allows for the exploitation of resources provided by large grassland areas (50% of agricultural area in the south of Belgium; [64]). On the other hand, the inclusion of some Belgian papers in Clusters 3 and 4 is justified by the intensification of dairy production that induces a decline of grazing practices [65]. The management of large herds (On average 273 cows/herd) following Dairy Australia coupled with climate issues are reflected in Australian studies [21,48,63,66,67]. The Australian production goal is higher than in NZ (5,731 L/cow per year vs. 4,235 L in NZ), requiring large feed supply. Grazed grass is thus complemented generally by cereals or concentrates and explains the inclusion of Australian studies in Cluster 3 [68]. Grass availability in NZ and Ireland allows the use of grass-based systems. Lower production levels of these systems were compensated by a higher number of cows. The profitability is linked to the productivity of the system rather than on individual cow production. Lowered feeding costs at high levels of grazed grass comply with other studies [6,69]. However, their decrease is more pronounced in Cluster 2, because of higher MY/cow per day coupled with high grass consumption.

Different factors reported in the clusters’ descriptions and published in selected studies were analysed with regards to milking frequency. The heterogeneity of data was noted in some comparisons but distribution in subgroups allowed to understand variation factors and to draw a general trend. Following data analysis, several parameters could be modified to increase milking frequency. Increasing the concentrate supply tended to raise milking frequency, but this effect was limited. In actuality, augmented CS at enlarged MMI (≥ 6 h) induced the only significant increase of this analysis. The analysis based on cluster allocation demonstrated results of similar size. Cluster 4 included only one study which aimed to compare high and low rumination levels of cows. This showed large individual cow variations; thus, the results would not be representative for this cluster. The effect of CS was not investigated in other studies of Clusters 3 and 4. However, reviewed papers from these clusters showed that CS fluctuated from 3.3 [70] to 7.5 kg in association with milking frequency ranges from 1.8 [70] to 2.47 milkings/cow per day [71]). It seemed to indicate that the relationship between concentrates and MF is not linear.

Minimum milking interval was considered to play a determinant effect on milking frequency as Lyons et al. (2014) [10] hypothesized. Short MMI allowed an increase in MF irrespective of the amount of supplied concentrate. This increase was less marked at MMI 4 h. Moreover, short MMI was a discriminating factor between Clusters 1 and 2 (MMI Cluster 1= 11.0 ± 0.6 h and 6.3 ± 0.6 h in Cluster 2). In Clusters 3 and 4, MMI was usually set at 6 to 8 h and the effect of change in MMI was not investigated.

Early lactation increased milking frequency. This was confirmed by other studies not included in the dataset [30,42]. Only the results from Lyons et al., (2013_c_) [34] appeared to indicate that lower MF in late lactation cows could be prevented by constant concentrate supply in ABC pasture allocation.

On the other hand, PDMI > 17 kg DM/cow per day decreased the MF. This fact was also pointed out in studies conducted in full-grazing systems [39] and in systems combining grass and access to barn [3,38]. Some practices can be implemented to control PDMI, e.g., pasturing at varying sward height or adapting the grazing management. For example, strip-grazing allows matching of grass availability and cows’ needs. Several authors [34,69,71] evaluated in pastoral systems the division of pasture into three blocks (ABC system) and this practice was replicated in other countries like Ireland [22,24,27,43]. This system was tested in France [13]. However, the effect on MF observed in this study was difficult to interpret, as the pasture allocation was adapted at the same time as the MMI.

The results of this analysis confirm that higher milking frequency induces an increase in milk yield and increased the system’s productivity. However, we observed effects of variable intensities between subgroups. The early stage of lactation at a CS of less than 1 kg/cow per day led to the sharpest rise. The combination of SOL with relatively constant CS and ABC pasture design showed lower impact, therefore milking frequencies and milk yield remained quite stable over the lactation length. Following the performed analysis, increasing MF by 1 unit (for example from 1.5 to 2.5 milkings/cow per day) might increase the MY by 14.1 kg/cow per day. This value complies with results from comparison between clusters. From the description of clusters on basis of PCA, the difference in MF between Cluster 1 and 4 estimated at 1.1 milkings/cow per day induced an increase in MY of 12.1 kg/cow per day.

It could be envisaged that higher MY (kg/cow per day) induced higher MF. It is demonstrated that MY and MF were linked, especially in robotic systems where the possibility of increasing MF from 2 to 3 milkings/cow per day induced an increase in MY. However, several studies highlighted that returns to the robot, indoors or at pasture, relied on cow motivation. This motivation was almost linked to feed supply, i.e., concentrate or grass rather than to milk pressure in udder [9]. In this context, we hypothesized that the decrease in MY observed in pasture-based AMS could be due to lower MF. This hypothesis was confirmed by our results.

This study shows that increasing CS affects moderately MF. Conversely, the increase in MF and MY can be achieved when high CS (reaching 6.56 kg/cow per day) is associated with complementation with dry or ensiled forages or PMR provided at barn.

Following our results, the implementation of a system based on high PDMI (i.e., 90% grazed grass) should focus on the following parameters. In this system, the effect of concentrate allocation is moderate. A delta of 2 kg concentrate/cow per day leads to an increase of 0.12 milkings/cow per day and a rise of 1.88 kg milk/cow per day. The milk response (MR) can thus be estimated at 0.94 kg milk/kg concentrate, which is in the range of studies conducted by Reis and Combs (2000) and Bargo et al. (2002) [72,73], that estimated it from 0.86 to 0.96 kg milk/kg concentrate, respectively. This milk response was even lower in the study of Lessire et al. (2017_a_) [25] which reported an MR of 0.56 kg/kg concentrate over the whole grazing season. In actuality, MR depends on grass availability and composition and increases at low grass allocation.

Concentrate supply has to be considered from an economic point of view and be targeted regarding the stage of lactation or lactation number as highlighted in Lessire et al. (2017a) [25]. The MMI could be set at 6–7.5 h, as MMI ≤ 4 h has no effect on milking frequency. High PDMI is linked to a decrease in MF and thus in MY. Grazing strategies thus have to manage grass availability, e.g., by dividing pastures into three blocks and privileging strip grazing to motivate the cows to return to the AMS. In Clusters 3 and 4, grazed grass was considered more as a complement to concentrate and PMR that enables MY to remain quite stable at barn and at grazing [31]. Economically, a slight decrease in feeding costs was observed in proportions depending on the composition of the complement. Rations mainly composed of forages allowed for the most marked decrease of feeding costs in these systems. The choice of system depends on forage resources but it is also necessary to find a balance between high cow productivity based on diet composed of grass, concentrate and PMR and system productivity based on low feeding costs, i.e., based on grazed grass and high number of medium to low-producing cows. The first option could make farmers vulnerable to the volatility of milk price and of raw materials while the second option could lead to increased susceptibility toward climatic conditions.

## 5. Conclusions

The milking performances of pasture-based AMS vary depending on the management systems. This review describes four management systems and identifies the factors to take into account to increase the productivity within each one. The association of a specific model is correlated to geographical constraints, e.g., broad grazeable areas or intensification of dairy production. The use of large amounts of grazed grass (90% grazed grass) causes a decrease in MF and MY, but management based on low inputs seems profitable from an economical point of view. Nevertheless, it is not applicable in all contexts. If grass resources are not sufficient, higher feeding costs need to be compensated by higher milk yield/cow per day. The complementation of grazed grass by concentrate supplied at the AMS in addition to dry or ensiled forages or PMR provided at barn is thus necessary to maintain high milking frequency and high milk yield.

## Figures and Tables

**Figure 1 animals-10-00913-f001:**
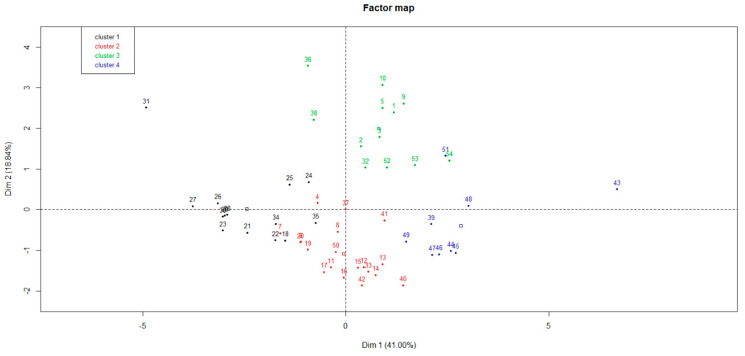
Graphic representation of clusters on axis 1 and 2, the numbers correspond to paper’s identifiers. Abbreviation: dim: dimension.

**Figure 2 animals-10-00913-f002:**
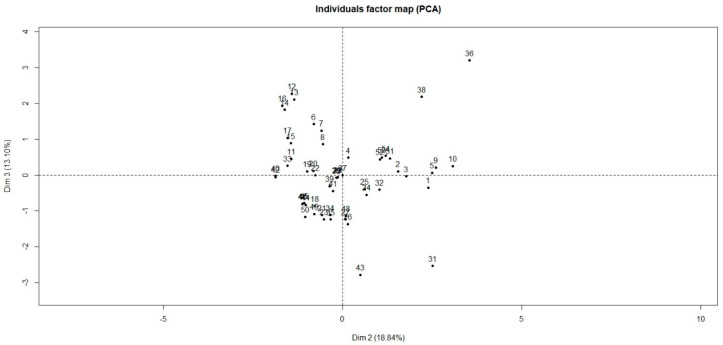
Graphic representation of clusters on axis 2 and 3, the numbers correspond to paper’s identifiers. Abbreviation: dim: dimension.

**Figure 3 animals-10-00913-f003:**
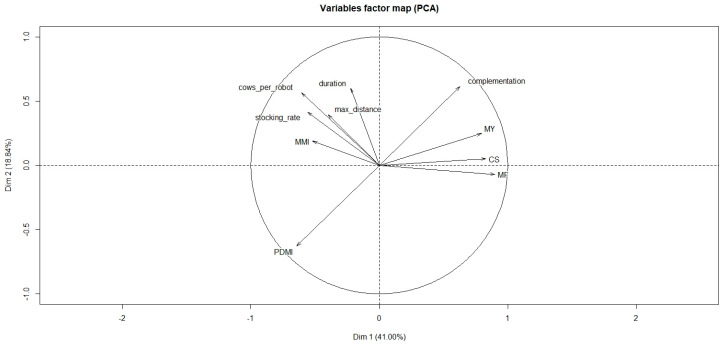
Projection and contribution of variables defining the clusters on the axes 1 and 2. Abbreviations: Dim: dimension; MMI: minimum milking interval (h); MY: milk yield (kg/cow per day); CS: concentrate supply (kg/cow per day); MF: milking frequency (milkings/cow per day); PDMI: pasture dry matter intake (kg DM).

**Figure 4 animals-10-00913-f004:**
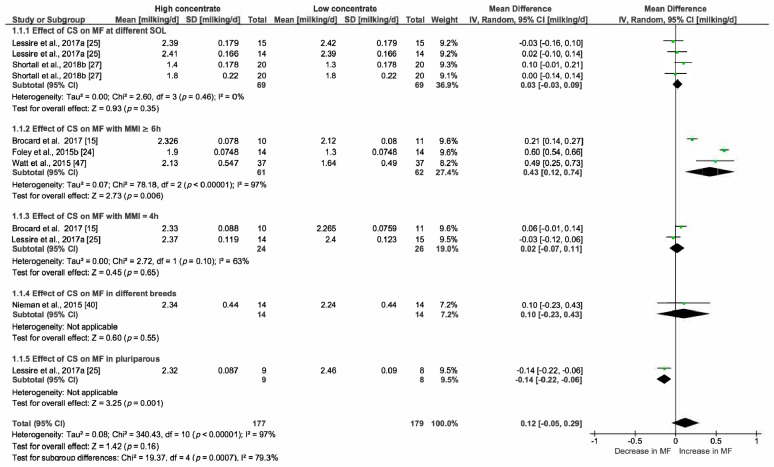
Forest plot of comparison: 1. Effect of concentrate supply (CS—kg/cow per day) on milking frequency (MF—milkings/cow per day), Abbreviations: SOL: stage of lactation (d); MMI: minimum milking interval (h).

**Figure 5 animals-10-00913-f005:**
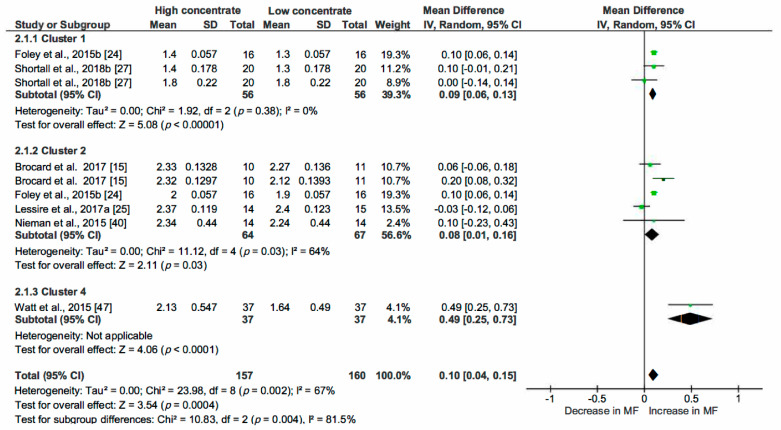
Forest plot of comparison: 2. Effect of concentrate supply (CS—kg/cow per day) on milking frequency (MF—milkings/cow per day) taking into account cluster allocation.

**Figure 6 animals-10-00913-f006:**
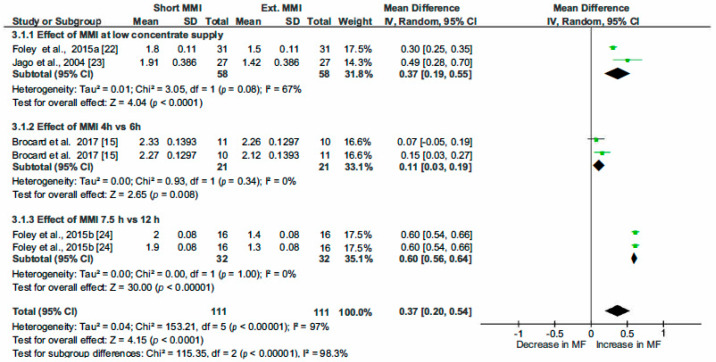
Forest plot of comparison: 3. Effect of minimum milking interval (MMI—h) on milking frequency (MF—milking/cow per day).

**Figure 7 animals-10-00913-f007:**
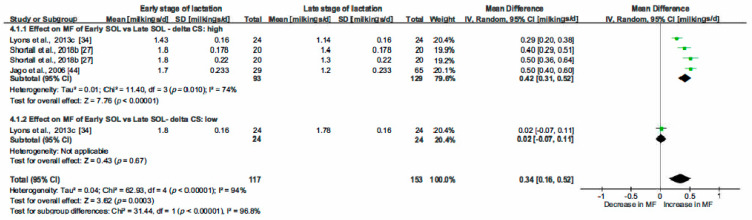
Forest plot of comparison: 4. Effect of stage of lactation (SOL—d) on milking frequency (MF—milking/cow per day). Abbreviations: CS: concentrate supply (kg/cow per day).

**Figure 8 animals-10-00913-f008:**
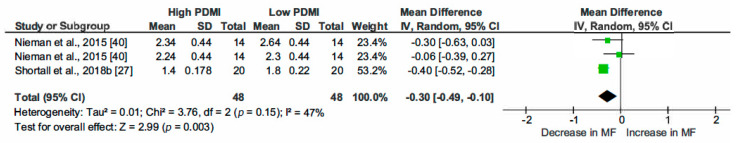
Forest plot of comparison: 5. Effect of Pasture Dry Matter Intake (PDMI—kg DM/cow per day) on milking frequency (MF—milking/cow per day).

**Figure 9 animals-10-00913-f009:**
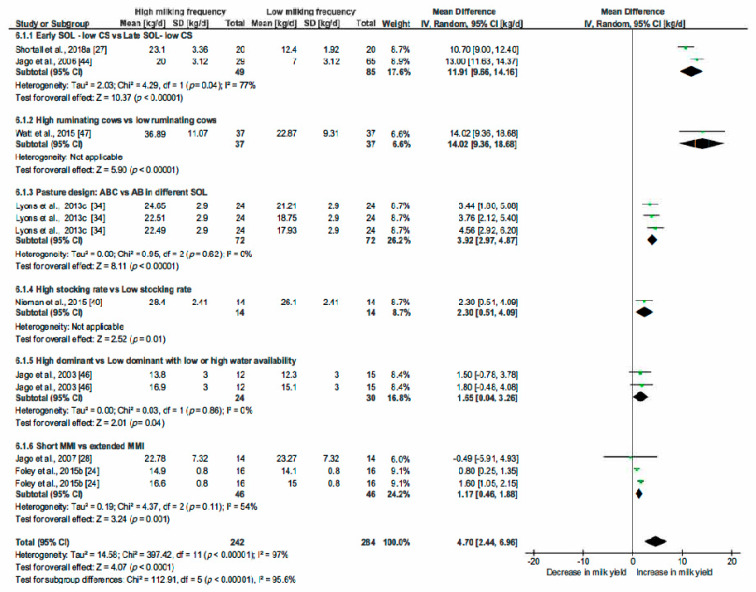
Forest plot of comparison: 6. Effect of milking frequency (MF—milking/cow per day) on milk yield (kg/cow per day). Abbreviations: CS: concentrate supply (kg/cow per day); SOL: stage of lactation (d); MMI: minimum milking interval (h).

**Table 1 animals-10-00913-t001:** Reviews included in the systematic review.

Category of Parameters	Reference	Objective of the Study	Reviewed Factors	Geographical Area of Studies
M	Brocard et al., 2017 [15]	Compilation of results EU project	Effect of MMI Effect of concentrate allocation	France Ireland Belgium Sweden
M – NM	Jacobs and Siegerford, 2012 [9]	Review General impact of AMS	Behaviour, Health, welfare AMS indoors-outdoors Pasture-based robot	Review NL Sweden NZ Australia
M	John et al. 2016 [16]	Review Optimising MF	Indoors/Outdoors 7 papers = pasture based	Australia Israël NL NZ
M	Kristensen et al., 2005 [17]	Review	Grassland management New technologies	DK Sweden NL
M	Lyons et al., 2014 [10]	Review	Optimizing MF of AMS combined with grazing Few hours Day Night and day	Australia NZ NL Sweden

Abbreviations: M: manageable; NM: non-manageable; AMS: automatic milking system; MF: milking frequency (milking/cow per day); MMI: minimum milking interval (h).

**Table 2 animals-10-00913-t002:** Classification of the selected research papers on basis of the effect of the studied factors on milking frequency.

Studied Factor								
Manageable /non-Manageable	Type of Manageable Factor	Description of Factor	First Author	Year	Objectives of the Study	Factor Improving MF	Lowering MF	Type of Publication
M	RM	Robot management	Davis et al. [18]	2008	Influence of washing time	No effect		Research
M			Davis et al. [19]	2005	Ways to improve efficiency	Decreased failed milking		Research
M			Munksgaard and Sǿndergaard [20]	2004	Managing practices	Forced vs. Free: no effect		Field reporting
M			Wildridge et al. [21]	2018_b_	Fetching at night	Fetching 23 h – 1 h	7d before	Research
M	RM	MMI	Foley et al. [22]	2015_a_	MMI	8 h	12 h	Proceedings
M			Jago et al. [23]	2004	MMI	6 h MMI	12 h MMI	Proceedings
M	RM	Concentrate allocation at the AMS	Foley et al. [24]	2015_b_	MMI X CS	8 h 3 kg	12 h 0.84	Research
M			Lessire et al. [25]	2017_a_	CS	4 kg = 2 kg		Research
M			John et al. [26]	2019_b_	Milking frequency	6.1 kg	4.9 kg	Research
M			Shortall et al. [27]	2018_a_	CS X SOL	Early lactation: 2.32 kg = 4.36 kg Late lactation: 0.42 kg = 2.42 kg		Research
M	RM	Concentrate allocation at the AMS X MMI	Foley et al. [24]	2015_b_	MMI X CS	8 h 3 kg	12 h 0.84 kg	Proceedings
M			Jago et al. [28]	2007	MMI CS	6 h 0 kg = 1 kg	8 h	Research
M			Brocard et al. [15]	2017	MMI X CS	4 h 4 kg	6 h 2 kg	Research
M	RM	Complementation at barn	Spörndly and Wredle [29]	2004	Distance Complementation	Dist 260 m + grass silage: no effect vs. Dist –no silage		Research
M			Lyons et al. [30]	2013_b_	Complementation	Post	Pre	Research
M			Lessire et al. [31]	2015_a_	Large herds	Grazing + PMR		Proceedings
M			Lessire et al. [32]	2017_b_	Large herds	Grazing + PMR		Proceedings
M	GM	Pasture allocation	John et al. [33]	2013	Pasture management	A < B and C	A > B and C	Field reporting
M			Lyons et al. [34]	2013_c_	Pasture allocation	ABC	AB	Research
M			Jago et al. [23]	2004	Pasture allocation	ABC	AB	Research
M			Cloet et al. [13]	2017	Pasture allocation	AB	ABC	Proceedings
M	GM	Pasture dry matter intake	Davis et al. [18]	2006	PDMI	Low pasture allowance		Proceedings
M			Jago et al. [35]	2010	Comparison 2 systems	Grass + CS (up to 3.7 kg)	Grass + 0.5 kg CS	Proceedings
M	GM	Pasture access	Huneau et al. [36]	2013	Access to pastures			Field reporting
M			Utsumi [37]	2011	Grass allocation	0 h grazing	12 h	Research
M			Van Dooren [3]	2004	Grass allocation	Day time grazing		Research
M	GM	Sward height	Ketelaar et al. [38]	2000	Sward height Distance	7.4 cm	11.4 cm	Research
	GM		Dufrasne et al. [39]	2012	Sward height	3.2 cm	10.2 cm	
M	GM	Distance to the AMS	Ketelaar et al. [38]	2000	Sward height Distance	No effect 146 m to 360 m		Research
M			Spörndly and Wredle [29]	2004	Distance	Dist = 50 m	Dist = 260 m	Research
M			Dufrasne et al. [39]	2012	Distance	No effect 100 to 425 m		Research
M	GM	Stocking rate	Nieman et al. [40]	2015	Stocking rate	No effect		Research
M	GM	Water allocation	Spörndly and Wredle [41]	2005	Water allocation	No effect of water for distance = 300m		Research
M	HM	Breed	Clark et al. [42]	2014_b_	Breed effect	No effect		Research
M	HM		Nieman et al. [40]	2015	Genetic	Ho		Research
M	HM		Shortall et al. [43]	2018_a_	Breed effect	No effect of breed (Ho vs. X Jersey vs. X Norw)		Research
M	HM	Stage of lactation	Jago et al. [44]	2006	Stage of lactation	19 d	266 d	Proceedings
M	HM		John et al. [26]	2019	Milking frequency	CS = 6.1 kg DIM 78.8d	CS = 4.9 kg DIM 104d	Research
	HM +RM		Lyons et al. [30]	2013_b_	Pre vs. post supplementation	No effect EL < 100d – Mid: 100-200d – late >200d		
M	HM +RM		Shortall et al. [27]	2018_b_	CS XSOL	19 ± 9 d	208 ± 9 d	Research
M	HM		Elischer et al. [45]	2015	Oxidative stress	DIM 21 DIM 7 = DIM 14	DIM 1	Research
M	HM	Parity	Elischer et al. [45]	2015	Oxidative stress	No effect Primi vs. Multi		Research
NM	X	Dominance	Jago et al. [46]	2003	Dominance	HD	LD	Research
NM	X	Rumination	Watt et al. [47]	2015	Rumination	HR	LR	Research
NM	X	Climate conditions	Wildridge et al. [48]	2018_a_	THI		THI > 68 d-1 d-2	Research
NM	X		Lessire et al. [49]	2015_b_	THI	Heat stress (THI = 70,5)	N (THI < 68)	Research

Abbreviations: M: manageable; NM: non-manageable; RM: robot management; GM: grazing management; HM: herd management; MF: milking frequency (milking/cow per day); MMI: minimum milking interval (h); PMR: partial mixed ration (kg DM); THI: temperature humidity index; HR: high ruminating cows; LR: low ruminating cows; HD: high dominance; LD: low dominance; d: days; DIM: days in milk; SOL: stage of lactation; ERL: early lactation; CS: concentrate supply (kg/cow per day); dist: distance (m); Ho: Holstein; X: crossed; Norw: Norwegian; Primi: primiparous; Multi: multiparous; PDMI: pasture dry matter intake (kg DM).

**Table 3 animals-10-00913-t003:** Values of the different parameters per cluster (Values are LSmeans ± SE). Values statistically different within columns are noted by different superscript, e.g: value in Cluster 1 marked with the superscript ^a^ is significantly different from Cluster 2 and 4 identified ^b^. The *p*-value of each tested parameter is indicated as is the R²-value.

Studied Parameter						
	Cluster 1	Cluster 2	Cluster 3	Cluster 4	*p*-Value	R²
**n studies**	**14**	**18**	**12**	**9**		
n cows per milking unit	75.1 ± 2^a^	51.3 ± 1.8^b^	67.1 ± 2.2^c^	48.5 ± 2.7^b^	<0.0001	0.69
Minimum Milking Interval (h)	11.0 ± 0.6^a^	6.3 ± 0.6^b^	6.9 ± 0.7^b^	7.8 ± 0.8^b^	<0.0001	0.47
Concentrate supply (kg/cow per day)	1.92 ± 0.35^a^	2.61 ± 0.32^a^	4.00 ± 0.38^b^	6.56 ± 0.42^c^	<0.0001	0.64
Complementation (kg DM/cow per day)	0.0 ± 0.7^a^	0.4 ± 0.7^a^	7.5 ± 0.8^b^	8.4 ± 1.4^b^	<0.0001	0.64
Milk yield (kg/cow per day)	16.8 ± 1.0^a^	20.4 ± 0.9^b^	27.0 ± 1.1^c^	28.9 ± 1.2^c^	<0.0001	0.64
Milking frequency (milkings/cow per day)	1.50 ± 0.08^a^	2.09 ± 0.07^b^	2.22 ± 0.08^b^	2.63 ± 0.09^c^	<0.0001	0.66
Pasture dry matter intake (kg DM/cow per day)	17.4 ± 0.8^a^	16.1± 0.7^a^	8.5 ± 0.9^b^	9.1 ± 1.4^b^	<0.0001	0.62
Stocking rate (n cows/ha)	3.07 ± 0.13^a^	2.16 ± 0.12^b^	2.92 ± 0.14^a^	2.01 ± 0.17^b^	<0.0001	0.47
Distance to the milking unit (m)	627 ± 35^a^	675 ± 30^a^	836 ± 38^b^	346 ± 44^c^	<0.0001	0.62
Duration experiment (d)	103 ± 31^a^	79 ± 28^a^	174 ± 34^a^	77 ± 39^a^	0.10	0.15

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
