# Peer review of "Systematic Review and Meta-Analysis: Identification of Factors Influencing Milking Frequency of Cows in Automatic Milking Systems Combined with Grazing"

_animals, 2020, doi:10.3390/ani10050913_

Round 1
Reviewer 1 Report
Brief Summary:
The main aim of the paper was to investigate the factors affecting milking frequency and then to examine impact of milking frequency on milk yield; this was done through statistical methods that allowed results from a large number of studies to be compiled and examined in this one study. The study (a) showed that the parameters influencing milking frequency included concentrate supplementation, milking interval, pasture intake and stage of lactation; (b) quantified the relationship between each parameter and milking frequency; and (c) identified the relationship between milking frequency and milk yield. Finally, by using and interpreting these results the study was able to incorporate an economic perspective and identify the effect of different scenarios on productivity and profitability, and what actions would be economically feasible if, e.g. one feed source was compromised.
Broad comments:
- In general, I would consider the paper to be very good.
- The question is original, is well defined and is very relevant. It is absolutely the case that there are numerous studies published on the impact of different factors on milking frequency and milk production, most are very individual to the geographical area and examine parameters/ effects that are relevant in that area. So it was very necessary to try to pull together all of this information and draw possible conclusions from it. This paper does that and the results provide an advance in current knowledge.
- A very good search of the literature appears to have been carried out and that is a key issue here.
- I am not a statistician but I am of the view that the results are arrived at correctly. A solid description of the methodology has been outlined and that methodology, or directions/decisions within it, was underpinned by consideration of the foregoing results, context and reasoning of a wide range of variables. I believe that the results are interpreted appropriately and conclusions justified. Hypotheses and speculations are identified properly.
- The paper is written well; even though the results section can be complex, it is written and presented in a user friendly manner that is easy to understand and follow. To me, the study is correctly designed and is technically sound. The data is sufficiently robust to draw the conclusions.
- The output is of interest to readers. The dairy sector and milking specifically accounts for a significant proportion of researchers and industry. Automatic milking is getting as important worldwide as conventional milking. This study has these features as well as access, nutrition, and economics aspects, so I think it is very relevant.
- There is definitely a benefit to publishing this work, such a study is needed. The work provides an advance in current knowledge in this area. The authors have addressed an important long-standing question through the use of appropriate statistical data analysis methodology.
- The English language is appropriate and understandable – but I have a number of suggested changes in the next section.Line 22: increases MF
- Line 14: It seems to us
Line 26: More and more dairy farms (up to more than one in four in some countries) are equipped
Line 29: usually causes
Line 39: allowed maintenance of MF and MY
Line 48: according to Barkema et al.
Line 53-54: on a scale………………………..milk yield – re-write
Line 55: AMS – use abbreviation
Line 56: allow maximum use of grass
Line 57: designed to allow
Line 59: that allows maintenance of a high milk yield
Line 68: could compensate for the lower income associated with a low milk production.
Line 93: had to be
Line 95: 43 papers from (or of) 71 were selected
Line 97: geographical area of the 43 studies was identified
Line 98: forages and a grazed grass allocation
Line 100: New Zealand and Ireland (16 and 11%,respectively), where grass was a major constituent of cows’ rations, the next quarter involved Belgian……………….
Line 101: 13 and 11%, respectively) with limited grazing seasons, and
Line 105: The type of
Line 106: the main proportion of ……….. The remainder comprised of three short
Line 108: refereed
Line 110: included
Line 113: aimed to
Line 114: statistical indicators
Line 115: of observations was considered very relevant
Line 116: accepted studies
Line 124: robot.
Line 126: managed
Table 1: May be better to un-justify text in table
Table 1: re-write NL-Sweden and NZ-Australia
Line 146: provided
Line 160: deduced
Line 172: 0.250 € - units
Line 174: at PMR was evaluated at
Line 175: concentrate feeding
Line 178: , respectively
Line 181: Descriptive analysis were carried out first and then analysis was completed in Multivariate
Line 188: allowed us to
Line 196: to ensure its objectivity.
Line 198: low, whereas above 50% it was considered as high.
Line 199: used random effect models to consider the variability
Line 205: Table 3
Line 220: Axis 1 and represented 41 and 19% of the total variation, respectively. Axis 3 achieved
Line 230: , respectively
Line 236: (negatively)
Line 238: Impact……………is this a heading or use :
Line 240: Methods Section, use same units of cost/kg or cost/100 kg Clusters
Line 241: 1 to 4, respectively
Line 248: were ?
Line 250: subgroups analysed when
Line 251: are indicated in Figures
Line 252: in the description of Results.
Line 256: cow traffic
Line 257: we hypothesized
Line 258: total number
Page of lines 259-288: (a) have consistency in units e.g. milkings/cow/day and milkings/cow per day; (b) use increase instead of rise – e.g. was effective in increasing; (c) be consistent with abbreviations e.g. concentrate supply is used here rather than CS; (d) should HC be high CS
Page of lines 289-310: should concentrate allocation be CS ? and again consistency of units. Line 306: was compared to an MMI and use increase rather than rise
Line 316: as opposed to
Line 318: in early compared to late lactation cows
Line 319: prompted us to
Line 321: late lactation cows
Line 323: 200 days received 4.65
Line 331 – consistency of units
Line 340: choose to challenge
Line 342: is 0.3 to 0.6 correct
Line 358: the effect of short……..vs long……..MMI (12 h) both of which showed ………..
358/359: summary, when increased MF induced increased MY, this effect
Page of lines 366-416: consistency in use of abbreviations, e.g. milking frequency and MF. Also re-write 375-377 (Yet to PMR). Also re-write 381-384 The lowest to 2 clusters).
Line 413: one study that compared high and low rumination levels of cows; this showed large individual cow variations, thus results would not be representative for this cluster.
Line 417: 71)]. It seemed to
Line 419: interval was considered to play
Line 420: irrespective of the amount
Line 425: appeared to indicate
Line 430: allows matching of
Line 432: and this practice was replicated in other
Line 446-448: re-write
Line 466: the choice of system
Line 473: vary depending on
Line 475: productivity within each.
Line 475: the association of a specific model is correlated with
Author Response
Letter to the first Reviewer
Dear Reviewer,
I thank you very much for your careful reading and for your constructive comments. Here you will find the corrections made in the text following your recommendations. All the changes are tracked in the newly submitted document.
Best regards,
Françoise
- The English language is appropriate and understandable – but I have a number of suggested changes in the next section.
- Line 22: increases MF OK
- Line 14: It seems to us changed
Line 26: More and more dairy farms (up to more than one in four in some countries) are equipped OK
Line 29: usually causes OK
Line 39: allowed maintenance of MF and MY OK
Line 48: according to Barkema et al. OK
Line 53-54: on a scale………………………..milk yield – re-write – Re-wroten
Line 55: AMS – use abbreviation OK
Line 56: allow maximum use of grass OK
Line 57: designed to allow OK
Line 59: that allows maintenance of a high milk yield OK
Line 68: could compensate for the lower income associated with a low milk production. OK
Line 93: had to be OK
Line 95: 43 papers from (or of) 71 were selected OK
Line 97: geographical area of the 43 studies was identified OK
Line 98: forages and a grazed grass allocation OK
Line 100: New Zealand and Ireland (16 and 11%,respectively), where grass was a major constituent of cows’ rations, the next quarter involved Belgian………………. Re-wrotten as suggested
Line 101: 13 and 11%, respectively) with limited grazing seasons, and OK
Line 105: The type of OK
Line 106: the main proportion of ……….. The remainder comprised of three short OK
Line 108: refereed OK
Line 110: included OK
Line 113: aimed to OK
Line 114: statistical indicators OK
Line 115: of observations was considered very relevant OK
Line 116: accepted studies OK
Line 124: robot. Not sure to have understand – replaced by AMS
Line 126: managed Do you mean to change manageable by managed?
Table 1: May be better to un-justify text in table OK
Table 1: re-write NL-Sweden and NZ-Australia OK
Line 146: provided OK
Line 160: deduced OK
Line 172: 0.250 € - units OK
Line 174: at PMR was evaluated at OK
Line 175: concentrate feeding OK
Line 178: , respectively OK
Line 181: Descriptive analysis were carried out first and then analysis was completed in Multivariate OK
Line 188: allowed us to OK
Line 196: to ensure its objectivity. OK
Line 198: low, whereas above 50% it was considered as high. OK
Line 199: used random effect models to consider the variability OK
Line 205: Table 3 OK
Line 220: Axis 1 and represented 41 and 19% of the total variation, respectively. Axis 3 achieved OK
Line 230: , respectively OK
Line 236: (negatively) OK
Line 238: Impact……………is this a heading or use : changed
Line 240: Methods Section, use same units of cost/kg or cost/100 kg Clusters OK
Line 241: 1 to 4, respectively OK
Line 248: were ? Changed – but not sure of what you meant
Line 250: subgroups analysed when OK
Line 251: are indicated in Figures OK
Line 252: in the description of Results. OK
Line 256: cow traffic OK
Line 257: we hypothesized OK
Line 258: total number OK
Page of lines 259-288: (a) have consistency in units e.g. milkings/cow/day and milkings/cow per day; (b) use increase instead of rise – e.g. was effective in increasing; (c) be consistent with abbreviations e.g. concentrate supply is used here rather than CS; (d) should HC be high CS – I checked
Page of lines 289-310: should concentrate allocation be CS ? and again consistency of units. Line 306: was compared to an MMI and use increase rather than rise I checked and modified
Line 316: as opposed to OK
Line 318: in early compared to late lactation cows OK
Line 319: prompted us to OK
Line 321: late lactation cows OK
Line 323: 200 days received 4.65 OK
Line 331 – consistency of units checked
Line 340: choose to challenge OK
Line 342: is 0.3 to 0.6 correct OK
Line 358: the effect of short……..vs long……..MMI (12 h) both of which showed ……….. OK
358/359: summary, when increased MF induced increased MY, this effect OK
Page of lines 366-416: consistency in use of abbreviations, e.g. milking frequency and MF. Also re-write 375-377 (Yet to PMR). Also re-write 381-384 The lowest to 2 clusters). I re-wrote and verified
Line 413: one study that compared high and low rumination levels of cows; this showed large individual cow variations, thus results would not be representative for this cluster. OK
Line 417: 71)]. It seemed to OK
Line 419: interval was considered to play OK
Line 420: irrespective of the amount OK
Line 425: appeared to indicate OK
Line 430: allows matching of OK
Line 432: and this practice was replicated in other OK
Line 446-448: re-write – I re-wrote it
Line 466: the choice of system OK
Line 473: vary depending on OK
Line 475: productivity within each. OK
Line 475: the association of a specific model is correlated with OK
Reviewer 2 Report
General Comments: As the use of robotic milking systems has increased, a review of factors affecting milking frequency and milk yield when using these systems in concert with pasture-based dairy production systems should contribute to the literature. However, this manuscript has a number of concerns that need to be addressed. First, it is very difficult to read. Part of problem seems to be the use of English and even the use of some words like exploitation or complement, but in some cases like the relationship between milking frequency and milk yield seem to be correlations that could go in either direction. In addition, while 43 papers were selected for use in the review, much fewer were used in the analysis of individual variables. The most extreme example of this was the use of only two studies in analysis of the seemingly important relationships between pasture dry matter intake and milking frequency. Furthermore, there seems to be little consideration of the effects of pasture forage quality as related to pasture botanical composition or pasture management which likely affects the response to supplemental grain or stored forages. Additional comments follow: |
|
Line |
|
14 |
Use of the word ‘us’ is unclear. |
15 |
Define ‘exploitation’ |
22 |
Change ‘rises’ to ‘increases’ |
23 |
While ‘The impact of MF on MY’ presumably means MF increases MY, why couldn’t the increasing MY increase MF? |
26 |
Omit ‘and more’ |
29 |
What is the effect of pasture-based AMS being compared to? |
37-38 |
While this effect seems obvious, was there actually an economic analysis conducted? At what feed prices would pasture-based dairies be justified? |
48 |
Remove ‘following’ and move the Barkema et al. (2015) reference to the end of the sentence. |
50 |
Change ‘grazing practices’ to ‘utilization of grazing’ |
56 |
Change to ‘ …exclusively on a diet of grass.’ |
61 |
Omit ‘Actually’ |
61-63 |
What is this compared to? |
71-72 |
The last sentence is unclear. |
75 |
The term ‘Prisma statement’ is unclear. |
95 |
Change ‘on’ to ‘of’ |
108 |
Change ‘referred’ to ‘refereed’ |
128-142 |
Tables 1 and 2 essentially present the raw data for the paper and really doesn’t add much to the results or discussion. Points in the data set that do add something to the analysis could be discussed in the text. |
151-155 |
Grazing management variables should include variables related to forage nutritional quality such as botanical and nutritional composition along with sward height and forage allowance. |
154 |
The term ‘complement’ is unclear. Should it be ‘mechanically harvested forage or feed’ or ‘stored forage or feed’? It would seem that supplemental grain could be considered a complement. |
214-215 and 220-223 |
It is unclear what Figure 1 and 2 represent. They should be more thoroughly discussed and described in text. |
228-229 |
Why isn’t complementation included as a variable in relation to axis 1 |
238 |
‘Impact of Developed Strategies on Feeding Costs’ is an unclear heading. |
Figure 3 |
Better define the Axis headings, Dim. |
296 |
Define ‘this parameter’. |
308-310 |
Change ‘rise’ to ‘an increase’. |
308-310 |
Wouldn’t this relationship be confounded with cows per robot? If too many cows were allotted per robot, it would seem that Minimum Milking Interval would have less impact. |
332-336 |
Only two studies for this important relationship? Also no consideration of pasture botanical or nutritional composition which would be very important. |
352 |
The ABC and AB grazing systems need to be fully described either here or in the methods. |
371 |
While the authors assume that increased milking frequency increases milk yield, what evidence is there that it isn’t milk yield increasing milking frequency. |
377 and 380 |
Should ‘grazed grass’ be changed to ‘grazed forage’ to include mixed pastures.? |
377-378 |
How was the proportion of grazed ‘grass’ in Clusters 2 and 4 determined? |
395 |
‘inclusion of some of them’ is unclear. |
398 |
The word ‘targeted’ is unclear. |
401 |
Should ‘to develop’ be ‘the use of’ |
401 |
Does the ‘Lower production levels’ refer to grazing-based systems? |
412-414 |
The relationship discussed in this sentence is unclear. |
427 |
What is high pasture dry matter intake being compared to? This would also seem to be an example where milk yield influences milking frequency. |
435 |
Is milking frequency controlling milk yield or vice versa? |
437 |
Change to ‘…at a concentration…’ |
441 |
Change ‘rise’ to ‘increase’ |
444-448 |
Unclear sentence |
448 |
Change ‘rising’ to ‘increasing’ |
450 |
It is unclear what parameters are being discussed. |
458-471 |
Very unclear. |
474 |
The word ‘exploitation’ is unclear. |
475 |
The phrase ‘The belonging to’ is unclear. |
476-478 |
What data is the conclusion related to the economic point of view are referring to? |
480 |
The word ‘complementation’ seems in unclear here as in other parts of manuscript, it seems to refer harvested forage or harvested forage and grain mixtures. Perhaps supplementation or substitution would be appropriate. |
Author Response
Letter to the 2d Reviewer
Dear Reviewer,
Thank you for your comments.
Hereafter you will find the answers and corrections made following your observations.
I sincerely hope that answers I am providing will satisfy you.
If any is incomplete or needs more explanation, please feel free to address me new questions.
Best regards,
Françoise Lessire
14 |
Use of the word ‘us’ is unclear. I removed “us” |
15 |
Define ‘exploitation’ « agricultural exploitations » - completed in the manuscript |
22 |
Change ‘rises’ to ‘increases’ OK |
23 |
While ‘The impact of MF on MY’ presumably means MF increases MY, why couldn’t the increasing MY increase MF? It is true that MY and MF are linked – especially in robotic systems, the possibility of increasing MF from 2 to 3 milkings/cow per day has induced an increase in MY. The concern with pasture-based robot is linked to decreased MY. The hypothesis was that decreased MY was due to lower MF reported in all the studies. Literature review shows also that feed is a more potent incentive for cows to be milked than milk pressure in the udder due to high milk yield (Jacobs et al., 2012). This point was argued in the paper. |
26 |
Omit ‘and more’ OK |
29 |
What is the effect of pasture-based AMS being compared to? Precised in the text |
37-38 |
While this effect seems obvious, was there actually an economic analysis conducted? At what feed prices would pasture-based dairies be justified? These points are discussed further in the text |
48 |
Remove ‘following’ and move the Barkema et al. (2015) reference to the end of the sentence. OK |
50 |
Change ‘grazing practices’ to ‘utilization of grazing’ OK |
56 |
Change to ‘ …exclusively on a diet of grass.’ OK |
61 |
Omit ‘Actually’ OK |
61-63 |
What is this compared to? completed |
71-72 |
The last sentence is unclear. changed |
75 |
The term ‘Prisma statement’ is unclear. More explanation about Prisma was provided |
95 |
Change ‘on’ to ‘of’ OK |
108 |
Change ‘referred’ to ‘refereed’ OK |
128-142 |
Tables 1 and 2 essentially present the raw data for the paper and really doesn’t add much to the results or discussion. Points in the data set that do add something to the analysis could be discussed in the text. These tables are included to allow the reader to find the information he is looking for within the results of the systematic review. It’s just informative. In systematic review and meta-analysis, it is strong recommended to give a summary about information gathered. |
151-155 |
Grazing management variables should include variables related to forage nutritional quality such as botanical and nutritional composition along with sward height and forage allowance. The aim of this article is to gather maximum information on several concerns that can be completed from the included references and aims to investigate factors that could influence MF. The dataset already included 14 parameters. Few missing data were recorded. As you highlighted, for pair-wise comparisons, few publications could be used because PDMI had to be mentioned low vs high in the same conditions to allow comparison. I recognize the importance of botanical and nutritional composition on grazing but data on these factors were not available in all the papers. Thus, including these in the dataset would lead to more incomplete data and so prevent analysis that could reach the objectives of the paper. Only one paper is about grazing of soybean (Clark et al., 2014). All the other papers are based on grazing pastures including a large part of grasses. |
154 |
The term ‘complement’ is unclear. Should it be ‘mechanically harvested forage or feed’ or ‘stored forage or feed’? It would seem that supplemental grain could be considered a complement. More precisions were given |
214-215 and 220-223 |
It is unclear what Figure 1 and 2 represent. They should be more thoroughly discussed and described in text. More precisions were provided |
228-229 |
Why isn’t complementation included as a variable in relation to axis 1: The value of this variable was completed |
238 |
‘Impact of Developed Strategies on Feeding Costs’ is an unclear heading. Corrected |
Figure 3 |
Better define the Axis headings, Dim. OK |
296 |
Define ‘this parameter’. Changed in “variables” |
308-310 |
Change ‘rise’ to ‘an increase’. OK |
308-310 |
Wouldn’t this relationship be confounded with cows per robot? If too many cows were allotted per robot, it would seem that Minimum Milking Interval would have less impact. In this pair-wise comparison, the Authors of the papers included divided the herd into 2 groups of similar size. Thus, one group out of the herd with a short MMI was challenged to another group with extended MMI. There is no confounding effect in the analysis as the groups presented the same number of cows. The MMI is a parameter that the farmer set by himself depending on his production objectives and of some other factors including the size of the herd. |
332-336 |
Only two studies for this important relationship? Also no consideration of pasture botanical or nutritional composition which would be very important. To be included in this comparison, the study has to give information on contrasted PDMI with their effect on MF. Only 2 studies provided these data. Concerning the systematic review, no publication studied the relationship between two botanical/ nutritional composition and their effect on MF. The systematic review included only one paper investigating soybean grazing (Clark et al., 2014). |
352 |
The ABC and AB grazing systems need to be fully described either here or in the methods. OK |
371 |
While the authors assume that increased milking frequency increases milk yield, what evidence is there that it isn’t milk yield increasing milking frequency. The concern you arose is discussed in “Discussion”.The aim of this research is about the lower MY observed at grazing compared with barn and its relationship with lower MF. Studies have demonstrated that milk pressure did not motivate cows to come back to AMS but grass or concentrate supply did (Jacobs et al., 2012). Literature review underlined that low MF in pasture-based AMS was observed. So, with regards with literature review, we assumed that low MF induced low MY and not the contrary. |
377 and 380 |
Should ‘grazed grass’ be changed to ‘grazed forage’ to include mixed pastures.? All the studies lead and included in this meta-analysis were on pastures composed of grass and legumes |
377-378 |
How was the proportion of grazed ‘grass’ in Clusters 2 and 4 determined? Explained in M&M |
395 |
‘inclusion of some of them’ is unclear. Re-wroten |
398 |
The word ‘targeted’ is unclear. Changed |
401 |
Should ‘to develop’ be ‘the use of’ OK |
401 |
Does the ‘Lower production levels’ refer to grazing-based systems? Yes and completed |
412-414 |
The relationship discussed in this sentence is unclear. Rewroten |
427 |
What is high pasture dry matter intake being compared to? This would also seem to be an example where milk yield influences milking frequency. High PDMI could be defined as 90% grazed grass in cow diet – it was specified in results |
435 |
Is milking frequency controlling milk yield or vice versa? This is discussed above as answer to your comments |
437 |
Change to ‘…at a concentration…’ OK |
441 |
Change ‘rise’ to ‘increase’ OK |
444-448 |
Unclear sentence Re-wroten |
448 |
Change ‘rising’ to ‘increasing’ OK |
450 |
It is unclear what parameters are being discussed. Re-wroten |
458-471 |
Very unclear. Re-wroten |
474 |
The word ‘exploitation’ is unclear. Term was changed into management |
475 |
The phrase ‘The belonging to’ is unclear. Re-wroten |
476-478 |
What data is the conclusion related to the economic point of view are referring to? Calculation of feeding costs was explained in M&M and results presented in 3.1.4. I extended the 3.1.4 to give more explanations. |
480 |
The word ‘complementation’ seems in unclear here as in other parts of manuscript, it seems to refer harvested forage or harvested forage and grain mixtures. Perhaps supplementation or substitution would be appropriate. Complementation was specified all around the manuscript |